# SWI/SNF Complex in Vascular Smooth Muscle Cells and Its Implications in Cardiovascular Pathologies

**DOI:** 10.3390/cells13020168

**Published:** 2024-01-16

**Authors:** Hongyu Liu, Yang Zhao, Guizhen Zhao, Yongjie Deng, Y. Eugene Chen, Jifeng Zhang

**Affiliations:** 1Department of Internal Medicine, Cardiovascular Center, University of Michigan Medical Center, 2800 Plymouth Road, Ann Arbor, MI 48109, USA; hongyl@umich.edu (H.L.); ypharan@umich.edu (Y.Z.);; 2Department of Molecular & Integrative Physiology, University of Michigan Medical Center, Ann Arbor, MI 48109, USA; 3Department of Vascular Surgery, The Second Xiangya Hospital, Central South University, Changsha 410011, China; 4Department of Cardiac Surgery, University of Michigan Medical Center, Ann Arbor, MI 48109, USA

**Keywords:** vascular smooth muscle cell, SWI/SNF complex, epigenetics, cardiovascular disease

## Abstract

Mature vascular smooth muscle cells (VSMC) exhibit a remarkable degree of plasticity, a characteristic that has intrigued cardiovascular researchers for decades. Recently, it has become increasingly evident that the chromatin remodeler SWItch/Sucrose Non-Fermentable (SWI/SNF) complex plays a pivotal role in orchestrating chromatin conformation, which is critical for gene regulation. In this review, we provide a summary of research related to the involvement of the SWI/SNF complexes in VSMC and cardiovascular diseases (CVD), integrating these discoveries into the current landscape of epigenetic and transcriptional regulation in VSMC. These novel discoveries shed light on our understanding of VSMC biology and pave the way for developing innovative therapeutic strategies in CVD treatment.

## 1. Introduction

Vascular smooth muscle cells (VSMCs) reside in the tunica media of vessels [1] and play critical roles in both vascular homeostasis and the pathogenesis of cardiovascular diseases (CVD). As early as the 1960s, it has been established that VSMC are enriched in atherosclerotic lesions in human specimens [2] and animal models [3,4]. The plasticity of VSMC has been identified as one of the major pathological factors in atherogenesis [5,6]. Key transcription factors and cofactors, like serum response factor (SRF) [7], myocardin [8], and Krüppel-like factor 4 (KLF4) [9], were identified during the investigation of SMC plasticity. Despite their essential role in DNA–protein interactions, the importance of chromatin remodelers in VSMC plasticity has often been overlooked. Summarizing the known and unknown of chromatin remodeling in VSMC research offers valuable insights into this understudied aspect. This review delves into the role of a primary ATP-dependent chromatin remodeler, the SWItch/Sucrose Non-Fermentable (SWI/SNF) complexes, in VSMC biology and its implications for cardiovascular diseases.

## 2. Roles of VSMCs from Physiology to Pathology

Like skeletal muscle cells and cardiomyocytes, the primary function of VSMCs is contraction. Under precise hormonal and neural control, VSMCs regulate blood distribution and blood pressure. To maintain the contractile phenotype, VSMC undergoes differentiation via the expression of a repertoire of contractile apparatus and regulators [10]. *Myosin heavy chain 11* (*MYH11*), also known as *smooth muscle myosin heavy chain* (*SMMHC*), is responsible for encoding SMC-specific myosin. Different from myosins in other muscle tissues, MYH11 lacks intrinsic ATPase activity. Instead, its activity is triggered when Ser 19 on the regulatory myosin light chain becomes phosphorylated. This phosphorylation is precisely controlled by myosin light chain kinase (MYLK) and myosin light chain phosphatase (MLCP). Thin myofilaments are composed of α-actin, also known as α-smooth muscle actin (α-SMA), which is encoded by *actin alpha 2* (*ACTA2*) [11,12]. Of note, variants of those genes have already been well-documented in patients with inherited aortic diseases [13,14]. This underscores the importance of fully functional contractile machinery in VSMCs for maintaining aortic health.

Another important role of VSMCs is to regulate vascular extracellular matrix (ECM) homeostasis through the production of components including elastin, collagen [15], and proteoglycans and regulation of ECM remodelers including matrix metalloproteinase (MMP) and tissue inhibitor of matrix metalloproteinases (TIMPs). These determine the vessel wall’s mechanical strength, compliance, and elastic recoil. As a result, ECM construction and maintenance of VSMC have emerged as central priorities in the field of bioengineering for blood vessel grafts [16].

Interestingly, the two major roles of VSMCs seem to be conflicting. During the initial embryonic development stage, VSMCs exhibit rapid proliferation [17], migrate, and actively secrete ECM components crucial for vasculogenesis. However, mature VSMCs lose those properties and become quiescent with a low proliferation rate and synthetic activity. In the mature aorta, VSMCs possess contractile apparatus and maintain mechanical strength. Nevertheless, during vascular injury, alterations in the environmental cues cause the loss of contractile markers, including MYH11 and α-SMA, coupled with increased proliferation, migration, and protein synthesis in VSMC. This type of dedifferentiated VSMC is essential for vascular repair. The transition between the contractile and synthetic states of VSMCs is termed “phenotypic switch”. It is thus logical to infer that the intrinsic plasticity of VSMCs is beneficial for adaption to complex environments and response to damage. However, substantial changes in diet and lifestyle may hijack this ability and disrupt vascular homeostasis.

## 3. Epigenetic and Transcriptional Regulation of VSMC Plasticity in Health and Diseases

Vascular smooth muscle cell plasticity in CVD, including atherosclerosis, has been well-documented [2]. Especially in the past decade, the emergence of new technologies, such as lineage tracing, single-cell RNA sequencing (scRNA-Seq), and spatial transcriptomics [18,19,20,21], has largely extended our knowledge of cell origin, characteristics, and transcriptomic profiles in the atherosclerotic lesions. Under pathological conditions, VSMCs transdifferentiate into various cell types such as foam cells [22,23], mesenchymal-stem-cell (MSC)-like cells [24], macrophage-like cells [25,26], adipocyte-like cells [27], osteochondrogenic cells [28,29], fibromyocytes [19]. Multiple reviews thoroughly discuss this phenomenon [5,30,31,32,33,34]. A dramatic shift in gene expression occurs during dedifferentiation or transdifferentiation. Such changes necessitate a sophisticated transcriptional regulation network in VSMCs to precisely control specific gene expression in response to environmental changes [34,35,36].

In differentiated VSMC, the expressions of contractile genes are governed by an array of transcription factors and coactivators [35,36,37] (Table 1; Figure 1 and Figure 2). In 1985, CC(A/T)_6_GG, also referred to as the CArG element, was identified in the promoter of the human cardiac actin gene [38]. Subsequent research has highlighted the crucial role of the CArG element in regulating muscle-specific genes [39]. Serum response factor (SRF) was then identified to bind with CArG elements and regulate downstream gene expression [7,40]. In addition, the discovery of myocardin [8] and its competition with Elk-1 for SRF interaction [41] revealed the delicate regulation of the VSMC phenotype.

Aside from myocardin/SRF, other transcription factors also play essential roles in regulating contractile genes [35]. The roles of transforming growth factor-β (TGFβ) in VSMC differentiation have been documented [79]. In 1997, the TGFβ control element (TCE), proximal to two CArG elements, was discovered in the promoter of *ACTA2* [80]. TGFβ increases contractile gene expression by facilitating the binding of SRF to the CArG elements, possibly through interactions between Smad3 and p300 [81]. Subsequent research on the synergetic function and direct interaction of Smad3 and myocardin has further elucidates the roles of TGFβ signaling in VSMC differentiation [48]. Another transcription factor, GATA binding protein 6 (GATA6), was found to be highly expressed in the VSMCs during development [82], and it was shown to protect against injury-induced VSMC phenotypic switch [50]. Further study has revealed that GATA6, NK3 homeobox 2 (NKX3-2), and SRF form a triad complex to regulate contractile gene expressions [52].

Conversely, in dedifferentiated VSMC, KLF4 plays an essential role [83]. The significance of KLF4 in VSMC was first discovered through yeast one-hybrid cloning against TCE [84]. Subsequent studies have firmly demonstrated that KLF4 was a potent repressor for VSMC differentiation [62,85]. In response to PDGF-BB treatment, Sp1, pELK1, and KLF4 cooperatively bind to the G/C repressor element flanked by two CArG elements in the promoters of contractile genes, including *MYH11* and *TAGLN*. In addition, KLF4 recruits HDACs, reducing histone acetylation in the promoters. Consequently, it diminishes SRF binding and the expression of contractile genes [86]. In the mouse atherosclerosis model, SMC-specific KLF4 knockout showed less mesenchymal-stem-cell- and macrophage-like cells derived from SMC, along with less lesion and increased fibrous cap thickness, underscoring significant roles of KLF4 in the cardiovascular diseases [9].

As we delve deeper into the research on the transcriptional regulation of contractile genes, epigenetic regulation gains increasing interest [87,88]. Structural investigations revealed that SRF does not bind to nucleosomal DNA [89,90,91], implying its exclusive binding to open DNA regions. Meanwhile, several factors affect the binding activity of SRF to the CArG elements [80,86,88]. These findings shed light on the significance of chromatin conformation in the transcriptional regulation of contractile genes. McDonald et al. discovered that H3K4me2, H3K79me2, H3K9Ac, and H4Ac are enriched in the CArG elements in the SMC but not in the non-SMC [86]. In addition, H3K4me2 tethers with SRF and myocardin, and this modification persists even after VSMC dedifferentiation [92]. Loss of H3K4me2 at the CArG elements causes a decrease in contractile gene expression through the reduced TET2 (ten-eleven translocation-2)-mediated DNA demethylation [77,93]. In contrast, in embryonic stem cells or other non-SMCs, H3K9me3 and H3K27me3 govern the repression of contractile gene expression [91]. Lysine demethylase 3A (KDM3A), previously known as JMJD1a, interacts with myocardin and demethylates H3K9me3, thus promoting the expression of contractile genes [69]. In addition, numerous histone modifiers have been identified to regulate contractile genes, including PR/SET domain 6 (PRDM6), SET and MYND domain containing 2 (SMYD2), SUV39H1, polycomb repressive complexes 2 (PRC2), and protein arginine methyltransferase 5 (PRMT5) (Table 1; Figure 1 and Figure 2) [68,71,72,73,74,75,78]. Although previous studies elucidated the intricate regulation of transcription and histone modification, the specific roles of chromatin remodelers in directly executing chromatin transformations require further investigation.

## 4. Chromatin Remodeling and SWI/SNF Complexes

In eukaryotic cells, DNA wraps around histones, forming the basic structural unit: a nucleosome [94]. In a rigid and delicate manner, DNA accessibility is highly regulated for complex activities, including replication, repair, and transcription. Chromatin remodeling involves altering interactions between histones and DNA, including assembly and disorganization of nucleosomes. Transcription requires DNA free from histones to interact with proteins such as polymerase II and transcription factors. Studies have shown that chromatin remodelers are essential in controlling pluripotency, cell fate, and differentiation [95,96,97]. Four prominent remodeler families have spiked the most study interest to date, including SWI/SNF, Imitation switch (ISWI), chromodomain helicase DNA-binding (CHD), and INOsitol requiring 80/SWI2/SNF2-Related 1 (INO80/SWR1) [98,99]. This review focuses on the SWI/SNF complexes.

The SWI/SNF chromatin remodeling complexes were first discovered in *Saccharomyces cerevisiae*. SWI stands for ‘switch’ as the relevant genes regulate the *HO* gene, which is crucial for mating type switching [100]. Similarly, SNF denotes ‘sucrose nonfermenting’ since these genes control the *SUC2* gene responsible for sucrose catabolism [101]. Subsequent research has found that some genes from these two screenings overlap, forming a complex that regulates chromatin structure [102]. SWI/SNF complexes use energy from ATP hydrolysis to mediate nucleosome sliding or ejection [103]. The mammalian SWI/SNF complex was purified in 1996 [104]. In mammals, either Brg1 or Brm serves as the ATPase of the complex; the SWI/SNF complex is also named the BRG1/BRM-associated factor (BAF). The initially isolated complex was termed canonical BAF (cBAF) [104]. Subsequent research has identified two more types of SWI/SNF assemblies: polybromo-associated BAF (PBAF) [105] and non-canonical BAF (ncBAF) [106,107,108]. These complexes are assembled by 9–16 subunits, and some subunits consist of several paralogs (Figure 3 and Table 2) [109]. Each subunit conveys unique functions for chromatin remodeling [110]. More than 1000 unique combinations of SWI/SNF subunits form diverse complexes with varied functions [111,112].

Recent advancements have significantly illuminated the structure of SWI/SNF complexes [113,114,115,116]. These findings further deepen our understanding of the chromatin remodeling mechanism of SWI/SNF complexes and the function of their subunits. According to Chen et al., SWI/SNF complexes can be divided into three modules: the motor module, actin-related protein (ARP) module, and substrate recruitment module (SRM) [117]. SRM can be further divided into nucleosome binding lobe (NBL), DNA binding lobe (DBL), and histone-tail binding lobe (HBL) [117]. The central motor module is the ATPase, SMARCA4 (BRG1), and SMARCA2 (BRM), which directly hydrolyze ATP to translocate DNA [118]. The ARP module comprises actin beta (ACTB), BCL7, and actin-like 6 (ACTL6). They serve the dual function of connecting ATPase with SRM and regulating the ATPase activity. Within the nucleosome binding lobe, SMARCB1, coupled with double PHD fingers (DPF) and SMARCC, attaches directly to the nucleosome acidic patch via the C-terminal domain (CTD) [119]. The DNA binding lobe interacts with extranucleosomal linker DNA and various factors. This lobe consists of several DNA binding domains like the HMG-box domain on SMARCE1 and polybromo 1 (PBRM1) [120], AT-rich interaction domain (ARID) on ARID1A/B, and ARID2 [121].

Of note, multiple interactors of SWI/SNF complexes exist to orchestrate gene transcription (some interactions validated by the GST pull-down assay are listed in Table 3). The SMARCD family, including SMARCD1, SMARCD2, and SMARCD3, are reported to mediate the interactions between the SWI/SNF complex and various factors, such as PPARG coactivator 1α (PGC1α) and CCAAT enhancer binding protein ε (CEBPε) [122,123]. Recently, Wolf et al. [124] fused TurboID [125] with SMARCD1 and revealed close associations of the SWI/SNF complex with various transcription factors and epigenetic machinery, including lysine-specific methyltransferase 2 (KMT2) family, nuclear receptor coactivator (NCOA) family and histone acetyltransferases (HATs). The SWI/SNF complex can also recognize histone modifications. The bromodomains within its subunit BRD7/9, SMARCA4, SMARCA2, and PBRM1 facilitate their binding to acetylated histones [126]. The chromodomains on the subunit SMARCC1 and SMARCC2 recognize methylated H3 [127]. Given their sophisticated structure and interactors, it is speculated that the whole complexes remodel the chromatin at precise locations and accurate time points.

## 5. SWI/SNF Complex in Cardiovascular Development

Coffin–Siris syndromes (CSSs) represent congenital disorders predominantly linked to mutations in the subunits of SWI/SNF complexes and are characterized by mental retardation [138]. *ARID1B* is responsible for CSS1, with *ARID1A* for CSS2, *SMARCB1* for CSS3, *SMARCA4* for CSS4, *SMARCE1* for CSS5, *ARID2* for CSS6, DPF2 for CSS7, *SMARCC2* for CSS8, *SMARCD1* for CSS11, and *BICRA* for CSS12. Recently, more phenotypes in the cardiovascular system have been noticed in CSS patients [139]. Among fetuses, 67% of patients present cardiac anomalies, and 53% of patients present vascular anomalies [140]. For instance, one case study has reported that one fetus with *SMARCC2* deficiency was diagnosed with tetralogy of Fallot, a rare congenital disease caused by a combination of four heart defects [141]. Considering the lethality of cardiovascular anomality and the higher possibility of termination due to poor prognosis, the actual incidence of cardiovascular complications in CSS might be underestimated.

Over decades of research on the mammalian SWI/SNF complex, numerous animal models have been developed. In the mouse model, *SMARCA4* null embryo dies at day three due to the arrest of differentiation [142], while *SMARCA2* knockout mice are viable [143]. This indicates that the roles of SMARCA4 and SMARCA2 differ in development. As an essential subunit in the SWI/SNF complex, DPF3 is highly expressed in the heart and skeletal muscle and serves as a histone modification reader. DPF3 possesses 2 plant homeodomains (PHDs), which facilitate its binding with acetylated H3 and H4, and H3K4me1/2. Knockdown *dpf3* in the zebrafish leads to irregular cardiac morphology and muscular fiber disarray [144]. Constitutive knockout of *BIRCA* (*BRD4 interacting chromatin remodeling complex associated protein*) (also known as *GLTSCR1* (*glioma tumor suppressor candidate region gene 1*)) results in embryonic lethality and cardiac defects, including ventricular septal defect, double outlet right ventricle and a thinner ventricular wall [145]. These results indicate the indispensable roles of the SWI/SNF complex in the development of the cardiovascular system.

## 6. SMARCA4 and SMARCA2 in VSMC Biology and Cardiovascular Diseases

SMARCA4 and SMARCA2 (also called Brg1 and Brm, respectively) are the two mutually exclusive ATPase subunits of the SWI/SNF complex, which hydrolyze ATP to provide energy for the complex to remodel chromatin. Emerging evidence from early cardiovascular studies illuminates the potential significance of the SMARCA family in VSMC. In 2003, Chang et al. [54] found that SMC-enriched protein cysteine and glycine-rich protein (CSRP) 1/2 bridge SRF with GATA4/5/6 to promote SMC contractile protein expression in pluripotent 10T1/2 fibroblasts, while skeletal-muscle-enriched CSRP3 inhibits this transition to VSMC. Further study in 2007 [55] found that *CSRP2* overexpression in adult cardiomyocytes promotes SMC-specific marker gene expression. At VSMC marker promoters, such as *MYH11* and *Calponin* (*CNN1*), an interplay of SRF, p300, and varying histone modifications was observed. Notably, SMARCA4 and SMARCB1, but not SMARCA2, were recruited, underlining the significance of the SWI/SNF complex in SMC marker gene expression in adult cardiomyocytes. Interestingly, the interaction between CSRP and the SWI/SNF complex is also captured via SMARCD1-TurboID in the abovementioned study [124].

Following Chang’s work, Herring and his colleagues conducted a series of studies investigating the roles of SMARCA4 in VSMCs. They employed a special cell line called B22 cells derived from NIH-3T3 fibroblast, which express a dominant-negative SMARCA4 (DN-SMARCA4) induced by tetracycline withdrawal [146]. In this system, the increase in contractile genes via overexpression of myocardin-related transcription factor (MRTFA) is attenuated by DN-SMARCA4. Notably, MRTFA promotes contractile protein expression in cervical cancer HeLa cells but not in adrenal carcinoma SW13 cells, as the latter lacks SMARCA4 or SMARCA2. When SMARCA4 or SMARCA2 is overexpressed in SW13, the ability of MRTFA to promote contractile protein expression is restored. Furthermore, DN-SMARCA4 led to a decrease in contractile protein expression in primary colon SMCs. Through in vitro investigations, it was proved that SMARCA4 directly binds with MRTFA but not with SRF. DN-SMARCA4 significantly interferes with SRF binding [42]. Similarly, SMARCA4 interacts with myocardin and permits its effect on contractile proteins [43]. SMARCA4 also regulates miR-143 and miR-145 expression in conjunction with myocardin or MRTFA. Only when SMARCA4 and myocardin or MRTFA are coexpressed in the SW13 cells can SRF be recruited to CArG elements in the promoter of miR-143/145 [147]. Among all the SRF targets, *ACTA2* is a special case that offers profound insight, as it appears to be less susceptible to the influence of SMARCA loss. *ACTA2* is already highly expressed in the B22 cells, and SRF binding at its promoters is much higher than that of MYLK and TAGLN. When MRTFA or myocardin is induced, SRF binding at the *ACTA2* promoter and *ACTA2* expression are increased. DN-SMARCA4 did not attenuate the effect of MRTFA or myocardin for *ACTA2*. This indicates a possibility that the *ACTA2* promoter in B22, fibroblasts, and SMCs remains accessible for SRF binding. The loss of SMARCA4 does not close this region, and MRTFA or myocardin can still effectively promote *ACTA2* expression.

Herring and his colleagues further specifically knocked out *Smarca4* with or without *Smarca2* in SMCs using SMMHC-Cre. About 33% of SMC-*Smarca4* knockout mice demonstrate cardiopulmonary abnormalities, including defects in the cardiac outflow tract, patent ductus arteriosus (PDA), and ventricular septal defect (VSD). Those defects are coupled with cyanosis and elevated neonatal mortality rates. In addition, striking phenotypes are noticed in the gastrointestinal (GI) tract. They found that SMC-*Smarca4*/*Smarca2* double knockout mice died within two weeks after birth, with dilated intestines. While SMC-*Smarca4* knockout mice consistently showed enlarged intestines, the SMC-*Smarca2* knockout mice demonstrated relatively fewer aberrations. Consistently, contractile proteins are significantly decreased in the SMC derived from surviving SMC-Smarca4 knockout and double knockout mice, including telokin, Myh11, Cnn1, Tagln, and Acta2. Moreover, SMCs exhibited increased apoptosis and decreased proliferation [148].

Additionally, compelling evidence suggests that SMARCA4 and SMARCA2 regulate VSMC proliferation and inflammation. Endothelin 1 (ET-1) enhances VSMC contraction, proliferation, and inflammation in VSMCs, playing multifaceted roles in CVD [149]. ET-1 was found to increase the expression of *SMARCA4* in rat aortic smooth muscle cells, while this can be attenuated by NaHS. Overexpression of *Smarca4* promotes VSMC proliferation and the expression of *Ntf3*, *Pcna*, and *Pdgfα*. Smarca4 binds at the promoters of those genes and modulates DNA accessibility [150]. SMARCA4 and SMARCA2 are critical for activating inflammatory genes, including *IL6*, *IL1β*, and *CCL2*, particularly when stimulated by ET-1 in VSMCs. A luciferase assay revealed that the binding of SMARCA4 and SMARCA2 at the promoters of those genes is increased in VSMC in response to ET-1 [151].

Additional evidence further indicates the roles of SMARCA4 in CVD. Several groups have reported an increase in SMARCA4 in the human standard type A aortic dissection specimens [152,153,154]. To mimic different types of heritable thoracic aortic aneurysm (TAA), two pathologic variants, TGFR2^G357W^ and ACTA2^R179H^, were, respectively, overexpressed in human aortic VSMC. Consistent with the patient’s specimen carrying that allele, contractile genes, including *smoothelin-1* (*SMTN1*), *CNN1*, *vinculin* (*VCL1*), and *Tagln*, are decreased, while HDAC9 is increased, especially in the nucleus. In this model, SMARCA4 is indispensable for HDAC9 and EZH2 binding at the promoters of contractile genes, which are then repressed [155]. In the ligation-induced carotid artery model, *SMARCA4* is also significantly increased in the adventitial SCA1^+^ smooth muscle cells (AdvSca1-SM). Oral administration of SMARCA4/SMARCA2 inhibitor PFI-3 can rescue ligation-induced vascular remodeling and inflammatory cell infiltration. Perivascular administration of SMARCA4 shRNA attenuates adventitial expansion and vascular fibrosis. Under normal or TGFβ-stimulated conditions, PFI-3 decreases Myh11, Cnn1, and Acta2 expression, indicating essential roles of SMARCA4/SMARCA2. The cleavage under targets and release using nuclease (CUT&RUN) assay reveals that, in the presence of PFI-3, the binding of SMARCA4 and the level of H3K27Ac at the *Acta2* promoter are reduced [156].

A prevailing controversy exists that SMARCA4 augments the expression of contractile proteins in vitro, but SMARCA4 increases with contraction protein decrease in human TAA samples. Since SMARCA4 and SMARCA2 constitute all SWI/SNF complexes, they might collaborate with diverse factors to activate and repress genes [157]. SMARCA may serve as a universal subunit for different regulations in the process of vascular diseases.

## 7. Roles of the SMARCD Family in VSMC

The human SMARCD family was first cloned and identified in 1996, with specific tissue distributions observed. Of note, SMARCD1 is universally distributed, SMARCD2 is predominantly expressed in the pancreas, and SMARCD3 is primarily found in the heart and skeletal muscle [158]. All three SWI/SNF complexes contain only one of the three SMARCD subunits: ncBAF only has SMARCD1, whereas cBAF and PBAF can contain any one of the three SMARCD members. SMARCD1-3 is one of the core subunits in the SWI/SNF complex. Knockout of SMARCDs disrupts the engagement of the ARID family and ATPase to the complex core [109].

The roles of the SMARCD family in myogenesis are extensively documented [159]. Smarcd3 participates in the early myogenesis in zebrafish [160]. Myoblast determination protein 1 (MyoD) stands as a pioneering transcription factor in myogenesis, initiating muscular cell differentiation, binding to closed chromatin, and aiding subsequent binding of other transcription factors. SMARCD3 and SMARCA4 interact with MyoD and bind at the promoter of myogenin, a muscle-specific transcription factor. The binding of MyoD at the promoter of Myogenin is facilitated by SMARCD3. SMARCD3 phosphorylation mediated by p38α is indispensable for SMARCA4 recruitment and muscle subsequent differentiation [161].

The roles of SMARCD3 in cardiovascular development were first reported in 2004 [162]. Bruneau and his colleague found that SMARCD3 is expressed in the early stages of cardiac development. *Smarcd3* KO causes abnormal cardiac development, including shortened outflow tract, hypoplastic atrium and ventricle, and left–right asymmetry [137,162]. In zebrafish, Smarcd3 cooperates with Gata5 (a functional homolog of mammalian Gata4) and T-box transcription factor 5 (Tbx5) to promote cardiac development [163]. Cardiac defects are found in both *Smarcd3* constitutive KO and cardiac-specific KO models driven by *Nkx2-5* or *Myh6* Cre [134].

SMARCD3 interacts with transcription factors to regulate cardiac developmental markers. Knockdown of Smarcd3 reduces natriuretic peptide A (NPPA), a cardiac differentiation marker. Additionally, NPPA–luciferase reporter is synergistically transactivated by TBX5, NKX2-5, GATA4, and SMARCD3, and this activation is blocked when *SMARCA4* is depleted. Moreover, the interaction between SMARCA4 and either TBX5 or NKX2-5 is critically dependent on SMARCD3. This aligns with other studies suggesting that SMARCD3 facilitates the interaction between transcription factors and the SWI/SNF complex [162]. Transient induction of *Smarcd3* along with *Tbx5*, *Nkx2-5*, and *Gata4* in the mouse embryo induces ectopic expression of cardiac marker *actin alpha cardiac muscle 1* (*Actc1*), *myosin light chain 2* (*Myl2*), *troponin T2* (*TNNT2*) in the mesoderm non-cardiac cells [164]. Furthermore, ectopic cardiac myocytes are found to be beating, which indicates the existence of a full contraction machinery. Notably, Gata4 and Smarcd3 interaction can induce the expression of *Actc1*, although the beating phenomenon is not observed. The addition of Tbx5 is required for the ectopic cardiac myocyte to beat. Moreover, Smarcd3 is essential for guiding Gata4 to its subsequent targets [164].

Although several studies have demonstrated the critical roles of SMARCD3 in developing skeletal muscle and cardiomyocytes, its contribution to VSMC biology was first reported in 2012. Sohni et al. [165] used rat multipotent adult progenitor cells (rMAPCs) as a platform. These cells exhibit the capability to differentiate into SMCs in MAPC basal medium supplemented with TGFβ1 and PDGFBB without the presence of Fetal bovine serum (FBS). During this differentiation process, the expression of *Smarcd3* and *Myocardin* is increased, along with the increase in contractile markers, including *Cnn1*, *Tagln*, *Smtn*, *Acta2*, and *Myh11*. They found that Smarcd3 is indispensable for the expression of *Tagln* and *Acta2*. Moreover, the elevation of Smarcd3 induced by TGFβ1 in rMAPC can be subdued by the Smad3-specific inhibitor and TGFβ receptor inhibitor. When the Smad3 binding element (SBE) upstream of Smarcd3 is mutated, TGFβ can no longer promote Smarcd3-luciferase activity, indicating that Smarcd3 is indeed regulated by TGFβ via Smad3 signaling. Moreover, Smarcd3 can interact with Srf and CArG elements in the promoters of *Acta2* and *Tagln*. This interaction indicates that SMARCD3 serves as a coactivator for SRF to promote the expression of contractile proteins.

Another SMARCD family member, SMARCD1, is also identified in the vascular system [166]. In primary rat VSMCs, free fatty acid (FFA, mixture of oleic acid, and palmitic acid) decreases *Smarcd1* in a dose- and time-dependent manner. With gain- and loss-of-function approaches, Smarcd1 was found to promote the expression of clock genes, including *Bmal1* (*basic helix-loop-helix ARNT like 1*), *Clock*, and *Dbp* (*D-box binding PAR bZIP transcription factor*). Notably, both overexpression and knockdown of *Smarcd1* only modulate the basal expression levels of these clock genes without affecting their circadian patterns. RORα has been identified to bind with Smarcd1, mediating the expression of *Bmal1*. Overexpression of *Smarcd1* results in increased levels of H3Ac and H3K4me3 and decreased H3K9me2 at the ROR-binding sites (RORE). Furthermore, overexpression of *SMARCD1* inhibits FFA-induced Erk phosphorylation and prevents VSMC proliferation and migration [166]. In 2020, we found that SMARCD1 is significantly upregulated in the human abdominal aortic aneurysm (AAA) specimen and AAA tissues from the mouse model [167]. SMC-specific *Smarcd1* knockout mice exhibit a dramatic decrease in AAA incidence and maximal diameter in both AAA models induced by elastase and Angiotensin II with hypercholesterolemia. Extracellular matrix (ECM) degradation and related proteases, including cathepsin S, matrix metalloproteinase-2 (MMP2), and MMP9, are significantly reduced in the Smarcd1^SMKO^ AAA specimen. We also observed reduced infiltration of Mac2^+^ macrophages and decreased levels of MCP-1 and IL-6 in the serum. Knockdown and overexpression of *SMARCD1* in the human aortic smooth muscle cell (HASMC) demonstrates that NF-κB downstream inflammatory genes, including *TNFα*, *IL1α*, and *IL1β*, are modulated by SMARCD1. ChIP-seq data showed reduced SMARCA4 binding at the TSS of NF-κB downstream genes, accompanied by decreased active histone markers, H3K9Ac and H3K27Ac. Interaction between p300 and SMARCA4 is also decreased when *SMARCD1* is knocked down.

Of note, SMARCD3 exhibits the opposite effects in AAA pathogenesis in mouse models [44]. Microarray data indicate that *SMARCD3* is significantly reduced in human AAA samples [168]. In the scRNAseq analysis of the mouse AAA model, Smarcd3 decreased specifically in VSMC [169]. SMC-specific Smarcd3 knockout (Smarcd3^SMKO^) mice showed higher AAA incidence and larger maximal diameters, accompanied by significant increases in ECM degradation, lymphocyte, and macrophage infiltration in the aortic wall [44]. Consistent with previous studies [165], SMARCD3 is essential for contractile gene expression in the VSMC. SRF and SMARCA4 binding at the promoters and H3K9Ac and H3K27Ac at the TSS of contractile genes are all regulated by SMARCD3. SMARCD3 is also required for interaction among SWI/SNF complex, SRF, and p300.

In addition, *SMARCD3* knockdown increases VSMC apoptosis [44]. Knockdown of *SMARCD3* decreases the expression of the anti-apoptotic gene *BCL2*. ChIP-seq and the ChIP assay demonstrate decreased SMARCA4, H3K9ac, and H3K27Ac binding at the KLF5 binding region within the promoter of *BCL2*. Knockdown of *SMARCD3* also reduces KLF5 binding at the promoter and its interaction with SMARCA4. In human pulmonary artery smooth muscle cells, prostate cancer cells, and colorectal cancer cells, KLF5 was found to serve as a protector against apoptosis, which induces antiapoptotic genes, including *BCL2*, *baculoviral IAP repeat containing 5* (*BIRC5*) and repress pro-apoptotic genes like *BCL2 associated X* (*BAX*) [170,171,172]. Our study uniquely highlights the pivotal role of SMARCD3 and the SWI/SNF complex in regulating VSMC apoptosis through KLF5 [44]. In alignment with our findings, SMARCD3 has been identified as a crucial factor for the survival of pancreatic ductal adenocarcinoma (PDAC) stem cells. We speculate that those cancer cells hijack the SMARCD3/KLF5 pathway to resist apoptosis [173].

## 8. Concluding Remarks and Future Directions

Research on VSMCs has evolved from studying physiological functions and contractile machinery to exploring transcription and epigenetic regulation. With its intricate plasticity mechanisms, this unique cell type likely holds the therapeutic potential for CVD.

The SWI/SNF complex has been identified as a target for an increasing number of drugs in cancer research [174]. For instance, Xiao et al. [175] found that androgen receptor (AR) and forkhead box A1 (FOXA1)-positive prostate cancer cells are sensitive to proteolysis-targeting chimera (PROTAC) degrader against SMARCA4 and SMARCA2. In VSMCs, however, SMARCA4 is essential for simultaneously expressing contractile proteins and inflammatory genes. Therefore, SMARCA4 and SMARCA2 are not ideal targets for cardiovascular disease treatment. Of note, the knockout of SMARCD1 in VSMC prevents inflammation and AAA formation [167]. Several studies have also highlighted the role of BRD9 in inflammation in β cells [176] and macrophages [177,178,179]. BRD9 is a unique subunit in ncBAF, while SMARCD1 is the only SMARCD family member in ncBAF. Since both BRD9 and SMARCD1 promote inflammation, we speculate that ncBAF plays an important role in inflammation in VSMC. Inhibitors targeting BRD9 or SMARCD1 hold promising potential for mitigating inflammation and addressing associated CVD.

It is crucial to comprehensively investigate the functions of specific SWI/SNF subunits or complexes in VSMC. There are three major complexes, encompassing 9–16 subunit families with over 1000 potential combinations. The diversity of the SWI/SNF complex determines the diversity of its targets and functions. As mentioned above, we have demonstrated the opposite functions of SMARCD1 and SMARCD3 in VSMC and AAA formation [44,167]. Contradicting functions between SMARCD1 and SMARCD3 have also been reported in skeletal muscle differentiation and metabolism regulation [180,181,182]. However, the role of the remaining family member, SMARCD2, in VSMC remains to be elucidated. It has been reported that SMARCD2, but not SMARCD1, partially functions as SMARCD3 in cardiomyocyte differentiation [164]. Interestingly, during in vitro differentiation of VSMC and cardiomyocyte, there is a decrease in SMARCD1 and SMARCD2, along with an increase in SMARCD3 [183,184]. It is still unknown whether SMARCD2 shares similar functions with SMARCD3 or SMARCD1 or possesses unique functions in VSMC biology. Further studies are essential for advancing our understanding of the intricate epigenetic regulation governing VSMC biology and associated vascular diseases.

## Figures and Tables

**Figure 1 cells-13-00168-f001:**
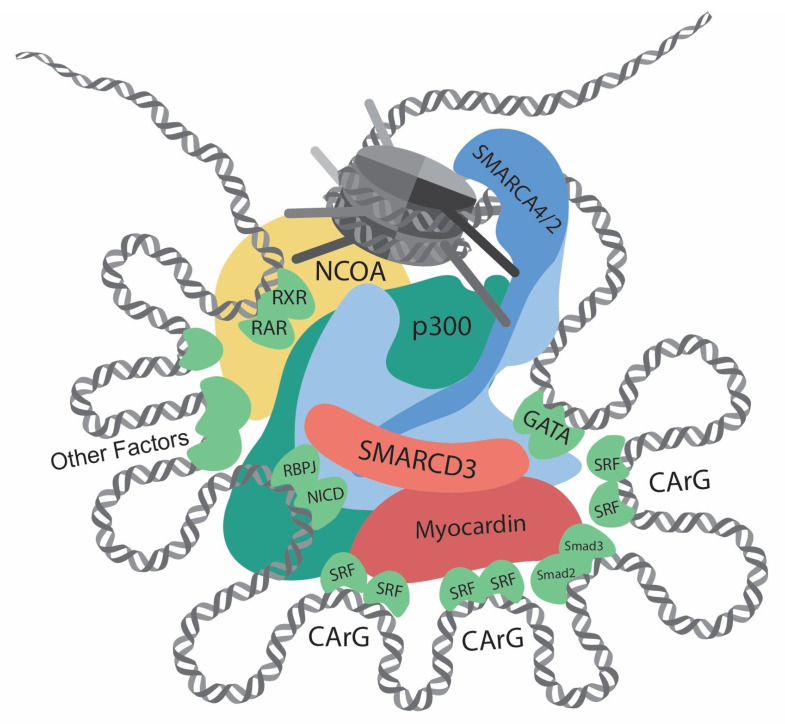
Transcriptional regulation at the CREs (cis-regulatory elements) of VSMC contractile genes. This diagram showcases the intricate network of interactions involving transcription regulators, such as chromatin remodelers, histone modifiers, transcription factors, and cofactors, at the CREs of the VSMC contractile genes. Chromatin remodelers that expose DNA regions by unwinding nucleosomes are central to the regulation. Several CArG elements, present at promoters and introns, bind to two SRFs and interact with myocardin. Within this region, there are binding sites for Smad2/3, NICD/RBPJ, and GATA. Retinoic acid signaling, crucial for VSMC differentiation, facilitates interactions between RAR/RXR and the SMARCD subunit of the SWI/SNF complexes. Other key contributors to VSMC differentiation include Prx1, Nkx3-2, PITX2, MEF2, and PIAS1. In addition, chromatin remodelers and transcription factors interact with histone modifiers, such as p300 and members of the NCOA family. Such complex interactions determine delicate and complicated regulation of contractile genes.

**Figure 2 cells-13-00168-f002:**
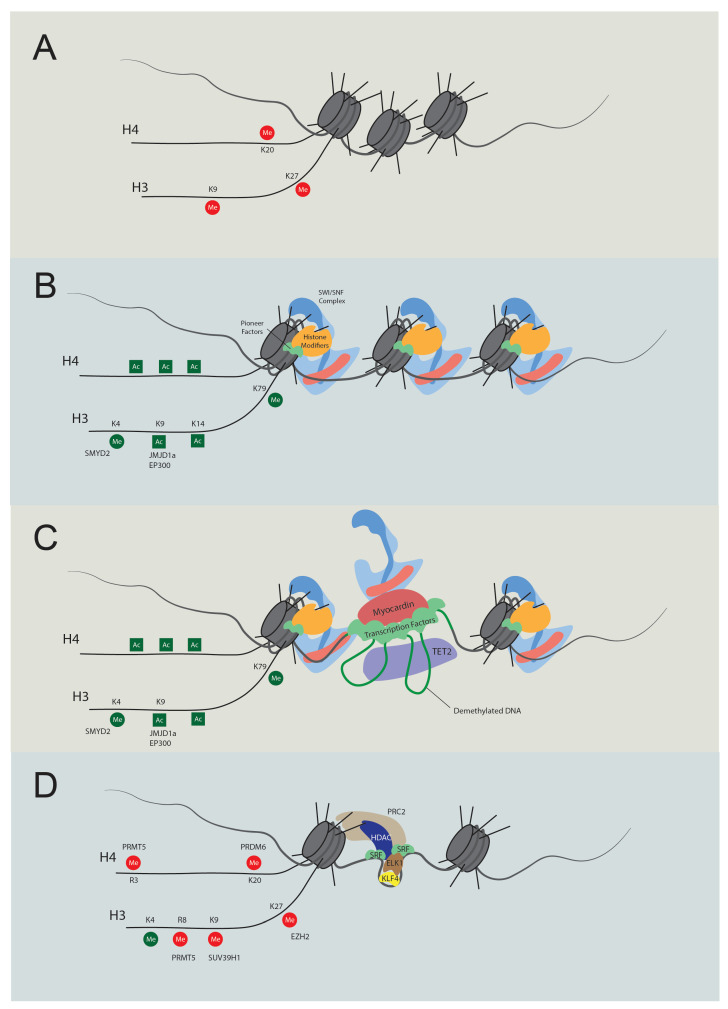
Temporal regulation of VSMC contractile gene expression. (**A**) Stem cell chromatin state: specific DNA regions are tightly wrapped into nucleosomes in stem cells. These regions are marked by repressive histone modifications, notably H4K20me3, H3K9me3, and H3K27me3, which play pivotal roles in maintaining gene silencing. (**B**) Activation of chromatin regions: during the transition to an active chromatin state, repressive histone marks are displaced, and active modifications emerge, such as H4Ac, H3K4Me2, H3K9Ac, H3K14Ac, and H3K79Me. Key players in this process include JMJD1a (H3K9me removal), SMYD2 (H3K4Me induction), and p300 (H3Ac and H4Ac induction). As histone acetylation weakens histone-DNA binding, pioneer factors can bind with nucleosomal DNA, subsequently recruiting either chromatin remodelers like the SWI/SNF complex or additional histone modifiers. (**C**) Transcription: the SWI/SNF complex introduces nucleosome-free regions, enabling TET2 to demethylate DNA. As a result, specific transcription factors bind to their respective elements. Subsequently, cofactors, including myocardin and CSRP, are recruited, promoting interactions between transcription regulators to promote transcription. (**D**) Dedifferentiation and transcriptional inactivation: in response to dedifferentiation signals like platelet-derived growth factor BB (PDGF-BB), the transcription of contractile genes was suppressed. In this situation, KLF4 binds to the G/C elements, while SRF partners with Elk1 rather than myocardin. Concurrently, HDACs and PRC are also recruited to deacetylate and induce methylation of H3K27. Additionally, PRDM6, PRMT5, and SUV39H1 mediate methylation at H4K20, H4R3 and H3R8, and H3K9, respectively. The loss of required transcription factors and DNA accessibility cease contractile gene transcription.

**Figure 3 cells-13-00168-f003:**
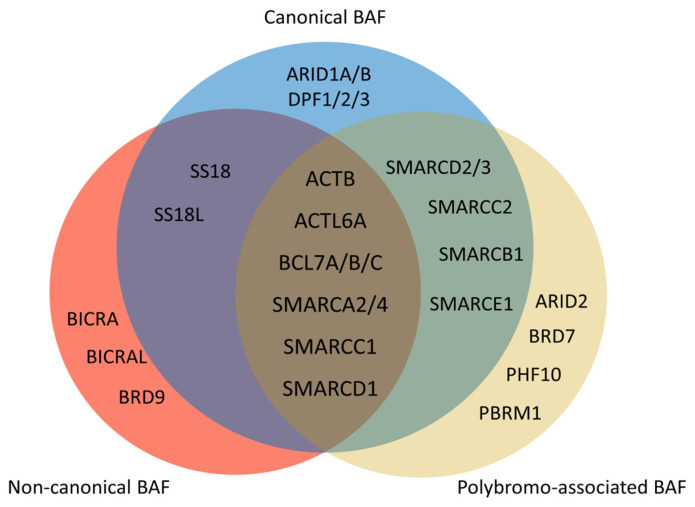
Composition of mammalian SWI/SNF complexes. The mammalian SWI/SNF chromatin remodeling complexes exist in three distinct types of assemblies: canonical BAF (cBAF), non-canonical BAF (ncBAF), and polybromo-associated BAF (PBAF). All three share several common subunits, such as the ATPase SMARCA2/4 and the ARP module (which includes ACTB, ACTL6A, and BCL7A/B/C). There are several differences between three types of assemblies. ARID1A/B and DPF1/2/3 only exist in cBAF. There are ARID2, BRD7, PHF10, and PBRM1 but no SS18/SS18L in PBAF. Non-canonical BAF possesses BICRA, BICRAL, and BRD9, and lacks SMARCC2, SMARCD2/3, SMARCB1, and SMARCE1.

**Table 1 cells-13-00168-t001:** Regulators of contractile gene transcription.

Gene	Category	Function	Effect on Contractile Gene Transcription	Reference
SMARCA4	Chromatin remodeler	Mediate chromatin accessibility	Dependent on interactors	[42,43]
SMARCD3	Chromatin remodeler	Mediate chromatin accessibility	Promotion	[44]
SRF	Transcription factor	Bind at CArG elements	Dependent on interactors	[8]
Myocardin	Cofactor	Interact with SRF	Promotion	[8,41,45]
MRTFA/B	Cofactor	Interact with SRF	Promotion	[41,46,47]
Smad2/3	Transcription factor	Bind DNA	Promotion	[48]
GATA4/6	Transcription factor	Bind DNA	Promotion	[49,50,51,52,53]
CSRP2	Cofactor	Interact with SRF, GATA6	Promotion	[54,55]
NKX3-2	Transcription factor	Bind DNA	Promotion	[52]
Prx1	Transcription factor	Bind DNA	Promotion	[56]
PITX2	Transcription factor	Bind DNA	Promotion	[57]
PIAS1	Transcription factor	Bind DNA	Promotion	[58]
MEF2	Transcription factor	Bind DNA	Promotion	[59]
Notch/RBPJ	Transcription factor	Bind DNA	Promotion	[60,61]
KLF4	Transcription factor	Bind G/C repressor element	Repression	[9,62,63,64,65]
Elk1	Cofactor	Interact with SRF	Repression	[47,63,64]
p300	Histone acetyltransferase	Increase histone acetylation	Promotion	[51,66,67]
HDAC	Histone deacetylase	Decrease Histone modification	Repression	[47,63,64]
SMYD2	Histone lysine methyltransferase	Increase H3K4me1, H3K4me3	Promotion	[68]
JMJD1A	Histone demethylase	Decrease H3K9me2	Promotion	[69]
WDR5	Cofactor	Increase H3K4me1, H3K4me3	Promotion	[70]
PRDM6	Histone lysine methyltransferase	Increase H4K20me2	Repression	[71,72]
SUV39H1	Histone lysine methyltransferase	Increase H3K9me3	Repression	[73]
EZH2	Histone lysine methyltransferase	Increase H3K27me3	Repression	[74,75]
TET2	Methylcytosine dioxygenase	DNA demethylation	Promotion	[76,77]
PRMT5	Histone arginine methyltransferase	H3R8me2, H4R3me2	Repression	[78]

**Table 2 cells-13-00168-t002:** SWI/SNF complex subunit HUGO name and common name.

HUGO Name	Common Name
SMARCC1	BAF155, SRG3
SMARCC2	BAF170
SMARCD1/2/3	BAF60A/B/C
SMARCB1	BAF47, INI1
SMARCE1	BAF57
ARID1A/B	BAF250A/B
ARID2	BAF200
PHF10	BAF45A
DPF1/2/3	BAF45B/D/C
BICRA/BICRAL	GLTSCR1/GLTSCR1L
SMARCA4	BRG1
SMARCA2	BRM
ACTL6A/B	BAF53A/B
PBRM1	BAF180

**Table 3 cells-13-00168-t003:** SWI/SNF subunit-interacting proteins validated by GST pull-down assay.

Interactor	SWI/SNF Subunit	Reference
AR	SMARCC1	[128]
CBP	SMARCA4	[129]
ERα	SMARCD1	[130]
ERα	SMARCD3	[131]
FOS	SMARCD1	[132]
JUN	SMARCD1	[132]
JUN	SMARCD3	[131]
MYC	SMARCA2	[133]
MYC	SMARCA4	[133]
MYC	SMARCB1	[133]
MYC	SMARCE1	[133]
Myocardin	SMARCD3	[134]
NCOA1	ARID1	[135]
NCOA1	SMARCC1	[128]
NCOA1	SMARCE1	[135,136]
Nkx2-5	SMARCD3	[134]
NR3C1/GR	SMARCD1	[130]
NR3C1/GR	SMARCE1	[130]
PCG1α	SMARCD1	[123]
PPARγ	SMARCD1	[123]
PPARγ	SMARCD3	[131]
PRMT5	SMARCB1	[133]
PRMT5	SMARCE1	[133]
RAR	SMARCD3	[135]
RBP-J	SMARCD3	[137]
RORα	SMARCD3	[131]
RXR	SMARCD3	[131,135]
SREBP1α	SMARCD3	[131]
Tbx5	SMARCD3	[134,137]

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
