# Peer review of "SWI/SNF Complex in Vascular Smooth Muscle Cells and Its Implications in Cardiovascular Pathologies"

_cells, 2024, doi:10.3390/cells13020168_

Round 1

Reviewer 1 Report

Comments and Suggestions for Authors

The review article summarized the functions of the chromatin remodeler SWI/SNF complexes in smooth muscle phenotype regulation and related cardiovascular development and diseases. The review is well written with proper figure illustration included. This reviewer has a few minor suggestions:

1.       In the Abstract and other sections, please spell out the full name of each abbreviation at its first use. E.g., CVD may be spelled out as cardiovascular diseases.

2.       The authors are suggested to carefully check grammar or each statement in the manuscript. E.g., Line 44-45, “VSMC undergoes differentiation via expression expressing a repertoire of contractile apparatus” could be changed to “VSMC undergoes differentiation via expression of a repertoire of contractile apparatus”.

3.       Line 60-61: “As a result, ECM construction and maintenance of VSMC have emerged as central priorities in the field of bioengineered blood vessel grafts”. Do the authors mean to say “As a result, ECM construction and maintenance of VSMC have emerged as central priorities in the field of bioengineering for blood vessel grafts”?

4.       Line 73: The transition between the contractile and synthetic states of VSMCs is termed the "phenotypic switch" may be changed to “The transition between the contractile and synthetic states of VSMCs is termed "phenotypic switch" or …is termed as "phenotypic switch".  

5.       Line 161: “These findings shed light to the significance of chromatin conformation…” may change to “These findings shed light on the significance of chromatin conformation…”

Comments on the Quality of English Language

The authors are suggested to carefully read through the manuscript to correct any potential grammar mistakes.  

Author Response

The review article summarized the functions of the chromatin remodeler SWI/SNF complexes in smooth muscle phenotype regulation and related cardiovascular development and diseases. The review is well written with proper figure illustration included. This reviewer has a few minor suggestions:

Response: We thank the reviewer for encouraging comments and edited the manuscript thoroughly according to your critiques.

  1. In the Abstract and other sections, please spell out the full name of each abbreviation at its first use. E.g., CVD may be spelled out as cardiovascular diseases.

Response: We made the changes accordingly.

  1. The authors are suggested to carefully check grammar or each statement in the manuscript. E.g., Line 44-45, “VSMC undergoes differentiation via expression expressing a repertoire of contractile apparatus” could be changed to “VSMC undergoes differentiation via expression of a repertoire of contractile apparatus”.

Response: Thanks for pointing out these errors. We made the changes and also revised it by a native speaker.

  1. Line 60-61: “As a result, ECM construction and maintenance of VSMC have emerged as central priorities in the field of bioengineered blood vessel grafts”. Do the authors mean to say “As a result, ECM construction and maintenance of VSMC have emerged as central priorities in the field of bioengineering for blood vessel grafts”?

Response: Thank you for the correction. We edited the manuscript accordingly.

  1. Line 73: The transition between the contractile and synthetic states of VSMCs is termed the "phenotypic switch" may be changed to “The transition between the contractile and synthetic states of VSMCs is termed "phenotypic switch" or …is termed as "phenotypic switch".  

Response: Thank you for the correction. We edited the manuscript accordingly.

  1. Line 161: “These findings shed light to the significance of chromatin conformation…” may change to “These findings shed light on the significance of chromatin conformation…”

Response: Thank you for pointing out this mistake. We made the changes accordingly.

Reviewer 2 Report

Comments and Suggestions for Authors

Abbreviations should be defined upon first mention, especially when a main focus of the work (i.e., SWI/SNF; CVD).

Page 1, lines 42-44: this sentence is confusing so please re-word.

Page 2, line 64: remove extra space.

Page 2, lines 67-68: this sentence is confusing so please re-word.

Page 2, line 79: proper grammar dictates not starting a sentence with an abbreviation.  If so desired, when starting a sentence spell out the full name of the word, and then use the abbreviation after that.

There are many instances of improper use, incorrect grammar and syntax, of the English language throughout the article.

There are several ‘over statements’ made in the work, as ‘we found’ and ‘our study highlights . . .’ yet no citations or references or supporting data are shown.  If definitive findings are being discussed, it would be prudent to include appropriate supporting references or absolute data in the article (OR cite it as ‘preliminary observations’).

The Concluding Remarks/Future Directions section should be a summary of the findings in the article proper, and new data and new discussion should not be included here (ie., discussion of the findings of Xiao et al [178], and many others here, should be in the main manuscript, not in this section).

Generally, this review article is well written and logical. 

Comments on the Quality of English Language

Abbreviations should be defined upon first mention, especially when a main focus of the work (i.e., SWI/SNF; CVD).

Page 1, lines 42-44: this sentence is confusing so please re-word.

Page 2, line 64: remove extra space.

Page 2, lines 67-68: this sentence is confusing so please re-word.

Page 2, line 79: proper grammar dictates not starting a sentence with an abbreviation.  If so desired, when starting a sentence spell out the full name of the word, and then use the abbreviation after that.

There are many instances of improper use, incorrect grammar and syntax, of the English language throughout the article.

There are several ‘over statements’ made in the work, as ‘we found’ and ‘our study highlights . . .’ yet no citations or references or supporting data are shown.  If definitive findings are being discussed, it would be prudent to include appropriate supporting references or absolute data in the article (OR cite it as ‘preliminary observations’).

The Concluding Remarks/Future Directions section should be a summary of the findings in the article proper, and new data and new discussion should not be included here (ie., discussion of the findings of Xiao et al [178], and many others here, should be in the main manuscript, not in this section).

Generally, this review article is well written and logical. 

Author Response

  1. Abbreviations should be defined upon first mention, especially when a main focus of the work (i.e., SWI/SNF; CVD).

Response: We thank the reviewer for pointing out this issue and make the changes accordingly.

      2. Page 1, lines 42-44: this sentence is confusing so please re-word.

Response: We rewrite the sentence to make it simple and clear. Thanks for the suggestion.

  1. Page 2, line 64: remove extra space.

Response: Done.

  1. Page 2, lines 67-68: this sentence is confusing so please re-word.

Response: Thanks for the suggestion, and we made the changes accordingly.

  1. Page 2, line 79: proper grammar dictates not starting a sentence with an abbreviation. If so desired, when starting a sentence spell out the full name of the word, and then use the abbreviation after that. There are many instances of improper use, incorrect grammar and syntax, of the English language throughout the article.

Response: We thank the reviewer for pointing out this issue and we edited the whole manuscript accordingly. The revised manuscript was also corrected by a native speaker.

  1. There are several ‘over statements’ made in the work, as ‘we found’ and ‘our study highlights . . .’ yet no citations or references or supporting data are shown. If definitive findings are being discussed, it would be prudent to include appropriate supporting references or absolute data in the article (OR cite it as ‘preliminary observations’).

Response: We thank the reviewer for pointing out this issue. We made the corrections and also cited the corresponding references.

  1. The Concluding Remarks/Future Directions section should be a summary of the findings in the article proper, and new data and new discussion should not be included here (ie., discussion of the findings of Xiao et al [178], and many others here, should be in the main manuscript, not in this section).

Response: We thank the reviewer for this insightful suggestion. We have implemented revisions in this section to succinctly encapsulate the role of the SWI/SNF complex in VSMC biology. Additionally, we have incorporated new information in this section pertaining to SWI/SNF complex-targeting drugs in other systems. They are used to support the future directions to develop innovative investigations and therapeutic strategies.

Reviewer 3 Report

Comments and Suggestions for Authors

This is a very interesting piece of work. 

In principal it would be good to have more display items to help the reader digesting this interesting article. 

page 2 line 45 sentence expression expressing

Comments on the Quality of English Language

Fine

Author Response

This is a very interesting piece of work. 

In principle it would be good to have more display items to help the reader digesting this interesting article. 

Response: We thank the reviewer for the encouraging comments. We made some changes to the figures and figure legends.

page 2 line 45 sentence expression expressing

Response: We thank the reviewer for pointing out this issue and we corrected it accordingly.